# Assessment of phytochemical screening, antibacterial, analgesic, and antipyretic potentials of *Litsea glutinosa* (L.) leaves extracts in a mice model

**Zubair Khalid Labu**[ORCID]*, **Samira Karim, Md. Tarekur Rahman, Md. Imran Hossain, Sarder Arifuzzaman**[ORCID] **Md. Shakil**

Department of Pharmacy, World University of Bangladesh, Dhaka, Bangladesh

* Labu1@pharmacy.wub.edu.bd

## Abstract

### Background

*Litsea glutinosa* (LG) leaves have been traditionally used in ethnomedicine for the treatment of various ailments, including pain, fever, and microbial infections. This study aims to scientifically evaluate the therapeutic potential of cold methanol extracts of LG leaves, specifically focusing on their analgesic, antipyretic, and antibacterial activities. In addition, the research includes preliminary phytochemical screening to identify key bioactive compounds and an acute toxicity test to assess the safety profile of the extract.

### Methods

In this study, we conducted an initial investigation of the major phytochemical groups present in *L. glutinosa* leaves using both modern chromatographic techniques, specifically High-Performance Liquid Chromatography (HPLC), and conventional phytochemical screening methods applied to cold methanol extracts. Both approaches consistently identified phenols and flavonoids as the predominant bioactive compounds. Following this phytochemical characterization, we assessed the analgesic efficacy of the extracts using acetic acid-induced writhing and electrical heat-induced nociceptive pain stimuli, evaluated antipyretic effects through Brewer's yeast-induced pyrexia, and determined antibacterial activity via the disc diffusion method. Additionally, the toxicity of the extracts was evaluated through preclinical testing.

### Results

In hot plate method, the highest pain inhibitory activity was found at a dose of 500 mg/kg of crude extract (3.37 ± 0.31 sec) which differed significantly (P < 0.01 and P < 0.001) with that of the standard drug morphine (6.47 ± 0.23 sec). The extract significantly prolonged reaction latency to thermal-induced pain in hotplate model. Analgesic activity at 500 mg/kg, LG extract produced a 70% suppression of writhing in mice, which was statistically significant (*p < 0.001*) compared to standard morphine's (77.5%) inhibition. In antipyretic

**Data availability statement:** All relevant data are within the paper.

**Funding:** The author(s) received no specific funding for this work.

activity assay, the crude extract showed notable reduction in body temperature (36.17 ± 0.32 °C) at dose of 300 mg/kg-body weight, when the standard (at dose 100 mg/kg-body weight) exerted (36.32 ± 0.67 °C) after 3 h of administration. In antibacterial studies, results showed that inhibition of bacterial growth at 400 µg dose of each extract clearly inhibited growth of bacteria from 11 to 22 mm. The extractives carbon tetrachloride fraction, chloroform soluble fraction, ethyl acetate fraction demonstrated notably greater inhibitory zone widths ($p < 0.05$) against tested strains.

## Conclusion

Overall, the cold methanol extract of LG leaves demonstrates the therapeutic potential in preclinical settings. Future research is warranted to isolate the specific bioactive compounds and elucidate their mechanisms of action to further support the development of new treatments and contributing to modern medicinal practices based on this plant leaves.

## 1. Introduction

In recent times, plant-based natural compounds have gained global prominence as complementary and alternative medicine, significantly contributing to the enhancement of health and well-being. Many widely used pharmaceuticals, such as aspirin, digoxin, morphine, ephedrine, quinine, tubocurarine, and reserpine, have their origins in medicinal plants. Phenols and flavonoids are particularly valued phytoconstituents due to their hydroxyl groups, which enable them to decompose peroxides, repair oxidative damage, and quench singlet and triplet oxygen [1]. The traditional knowledge of medicinal plants and their uses by indigenous cultures is not only vital for preserving cultural traditions and biodiversity but also essential for community healthcare and modern drug development. Synthetic tropical therapies often come with numerous side effects and are costly, making them inaccessible to many. To address this issue, plants available in the vicinity are often used without scientific validation. The use of higher plants and their extracts to treat infections is a long-established practice, with herbal medicines gaining popularity for their cost-effectiveness and eco-friendliness [2]. *L. glutinosa*, belonging to the Lauraceae family, is a well-known evergreen species found in the forests of Chittagong, Tangail and Sylhet districts in Bangladesh. It is occasionally cultivated in various parts of the country. The leaves, known for their mucilaginous properties, are used as antispasmodic, emollient, and poultice treatments. They are also employed in treating diarrhea, dysentery, wounds, and bruises [3]. An ethnopharmacological appraisal of *L. glutinosa* is justified due to its traditional medicinal uses, potential for bioactive compound discovery, the opportunity to fill scientific gaps, and the promotion of conservation and sustainable use. This research could pave the way for new therapeutic applications and strengthen the link between traditional knowledge and modern medicine. *L. glutinosa* is rich in bioactive compounds such as flavonoids, phenolic acids, tannins, terpenoids, alkaloids, coumarins, lignans which possess a wide range of therapeutic properties including antioxidant, anti-inflammatory, antimicrobial, and anticancer activities. The leaves have been reported for their use in treating the spontaneous and excessive flow of semen in young boys [4]. Additionally, the leaves extract exhibits antibacterial, analgesic and cardiovascular activities [5]. The berries produce oil that some tribal practitioners use to treat rheumatism. Common constituents of this species include tannin, β-sitosterol, and actinodaphnine, along with other compounds such as Boldine, norboldine, laurotetanine, n-methyllaurotetanine, n-methylactinodaphnine, quercetin, sebiferine, and litseferine [6].

Pain, an unpleasant sensory and emotional experience associated with actual or potential tissue damage, is relieved by analgesic compounds that act on the central nervous system or peripheral pain mechanisms without significantly altering consciousness. Analgesics are typically used when the noxious stimulus cannot be removed or as an adjunct to more etiological pain treatments [7,8]. Inflammation, the response of living tissues to injury, involves a complex array of enzyme activation, mediator release, fluid extravasation, cell migration, tissue breakdown, and repair. Non-steroidal anti-inflammatory drugs (NSAIDs) are frequently prescribed due to their effectiveness in treating pain, fever, inflammation, and rheumatic disorders. However, their use is linked to adverse effects on the digestive tract, ranging from dyspeptic symptoms, gastrointestinal erosions, and peptic ulcers to more severe complications such as bleeding or perforation [9]. Thus, developing new anti-inflammatory drugs with fewer side effects remains crucial, and natural products like medicinal plants could lead to discovering new, safer anti-inflammatory agents [10].

Pyrexia, or fever, often results from infections, tissue damage, inflammation, graft rejection, malignancy, or other diseased states. The body's natural defense mechanism raises the temperature to create an environment where infectious agents or damaged tissue cannot survive. Typically, infected or damaged tissue initiates the increased formation of pro-inflammatory mediators (cytokines like interleukin 1 β, α, β, and TNF-α), which enhance the synthesis of prostaglandin E2 (PGE2) near the preoptic hypothalamus area, triggering the hypothalamus to elevate body temperature [11]. Most antipyretic drugs prevent or inhibit COX-2 expression to reduce elevated body temperature by inhibiting PGE2 biosynthesis. However, synthetic agents irreversibly inhibit COX-2 with high selectivity, which can be toxic to hepatic cells, glomeruli, the brain cortex, and heart muscles, whereas natural COX-2 inhibitors usually have lower selectivity and fewer side effects [12]. *L. glutinosa* was chosen for this study due to its availability in Bangladesh and its traditional use in rural areas for various treatments. To date, no investigations have been conducted on this plant native to Bangladesh. Our primary aim was to evaluate the in antipyretic, analgesic activities and antibacterial properties of the plant leaves to validate their traditional uses.

## 2. Materials and methods

### 2.1. Sampling and proper documentation

Plant leaves were collected from Mirzapur village in the Tangail district of Bangladesh (geographical coordinates: 24.05º N, 89.92º E), July 2023. where the plant leaves were collected. After completing the identification of plant leaves by Bangladesh National Herbarium's taxonomist followed by provided taxonomical accession number # 354079 and with all necessary permissions obtained to continue our study.

### 2.2. Chemicals and reagents

Analytical-grade chemicals were used throughout the study, ensuring accuracy and consistency. The following reagents were employed:

- Methanol (liquid chromatography grade, ≥99.8%)

- Ethyl acetate (≥ 99.9% GC)

- Petroleum ether (≥ 80%)

- Chloroform (≥ 99% ACS Reagent Grade)

- Carbon tetrachloride (≥ 99.9%)

- Gallic acid (98%)

- Catechin (≥ 99.8%)

- Folin-Ciocalteu reagent (standard reagent grade)

- Aluminum chloride (anhydrous, sublimed, ≥ 99.8%)

All reagents were sourced from Science Park Chemicals Ltd., Bangladesh.

### 2.3. Equipment and instrumentation

To ensure precision and minimize experimental errors, all instruments were calibrated according to standard procedures. The equipment used in the study included:

- UV-Vis spectrophotometer (Model: UV-1700 series)

- Casio digital stopwatch (Model: HS-70W-1DF)

- Mechanical dryer (Model: CG-CG23KW-150KW)

- Digital analytical balance (Model: PS.P3.310)

- Thermostat water bath (Model: HHW21.420AII)

- Autoclave (Model: DSX-280KB)

- Rotary evaporator (Model: PGB002)

- Vortex mixer (Model: VM-11)

- Sonicator (Model: ULP-3000)

- High-performance liquid chromatography (HPLC) system (Model: LC-2060 3D, Shimadzu)

### 2.4. Drying and grinding of plant leaves

After removing extraneous leaves and undesirable materials, the collected plant leaves were thoroughly washed with water to eliminate any residual soil or debris. The leaves were then air-dried under sunlight for one week, followed by further drying using a mechanical dryer. Prior to analysis, the dried leaves were mechanically ground into a coarse powder, stored in an airtight container, and kept in a cool, dry, and dark environment to ensure optimal preservation and prevent degradation.

### 2.5. Plant material extraction

A total of 400 grams of the substance, in powder form, was placed in a clean, amber glass container with a flat bottom. The powder was submerged in 1500 mL of methanol at a temperature of 25 °C. The container was tightly sealed to prevent air from entering and stored for ten days. During this period, the contents were intermittently shaken and stirred to enhance the extraction process.

After the extraction period, the mixture was decanted and filtered through cotton, followed by filtration using Whatman No. 1 filter paper. The filtered extract was concentrated using a rotary evaporator at 40 °C [13]. This process yielded a sticky, greenish-black residue, which was identified as the crude methanolic extract.

Eleven grams of the crude methanolic extract were dissolved in 90% methanol using the Kupchan technique, and the mixture was then dissolved stepwise in different organic solvents with different polarity to get different fractions of crude methanolic extract [14]. Upon

complete evaporation of the solvent, the following fractions were collected: Ethyl acetate fraction (ESF, 3grams), Petroleum ether fraction (PSF, 1 grams), Chloroform fraction (CSF, 3 grams), Carbon tetrachloride fraction (CTF, 1.2 grams), Methanol soluble fraction (MSF, 2 grams), Aqueous fraction (AQF, 0.8 grams). All fractions were stored in a cold environment for subsequent biological analysis.

## 2.6. Phytochemical screenings

The confirmatory qualitative phytochemical screening of the crude extracts was conducted to detect the major classes of compounds, including tannins, saponins, flavonoids, alkaloids, phenols, glycosides, steroids, reducing sugars, and carbohydrates. Standard protocols were followed for this analysis [15].

The crude extract fractions obtained were assessed to identify the presence of glycosides, resins, phenols, alkaloids, saponins, steroids, and flavonoids. Although earlier studies have reported the presence of various phytochemicals in *L. glutinosa* and related species, it was deemed necessary to re-confirm their presence in the current samples. Environmental conditions, soil composition, and geographical location can affect the concentration and occurrence of certain phytochemicals, making re-screening critical for validation.

The re-screening process not only verified previously reported compounds but also allowed for the potential detection of minor or overlooked compounds that could contribute to biological activity. This comprehensive screening ensures that any biological or therapeutic effects are accurately associated with the phytochemicals present in the sample from the specific collection site. The findings from this phytochemical analysis will serve as the foundation for subsequent quantitative analyses and biological evaluations.

## 2.7. High performance liquid chromatography (HPLC)

HPLC stands as the well-known method for the chemical standardization of plant extracts. In this research study, gallic acid served as the standard for phenolic compound identification, while catechin functioned as the standard for identifying flavonoid compounds. The HPLC analysis employed a LC-2060 3D HPLC system, a Shimadzu model from Japan. The C18 column (ID: 5-micron x 100 Å) was used as the stationary phase, which gives a larger surface area the mobile phase has to travel across. The HPLC system incorporated a data detector A - Ch1. The mobile phase, operating in an isocratic gradient system, comprised 30% acetonitrile and 70% water. Before use, the mobile phase underwent filtration through a 0.45 μm filter paper and degas by sonication. The flow rate was selected at 1 mL/min. In this case, methanol was adjusted to 0.75 mL/min and distilled water was adjusted to 0.25 mL/min and the HPLC column was maintained at a temperature of 45 °C. Sample injection involved a volume of 20 μL, run time for the sample of 10 min and standard 5 min [16].

## 2.8. Determination of total phenolic content (TPC)

With minor adjustments, the standard methodology was applied to determine the Total Phenolic Content (TPC) using the Folin-Ciocalteu reagent [17]. Two milliliters of the crude methanolic extract and its organic-soluble components were mixed with 2 mL of distilled water to achieve a final concentration of 1 mg/mL. A 0.5 mL aliquot of this extract (1 mg/mL) was combined with 2 mL of Folin-Ciocalteu reagent, previously diluted 10-fold with deionized water, in a test tube. The mixture was left to stand at room temperature (22 ± 2 °C) for 5 minutes, followed by the addition of 2.5 mL of 7.5% sodium carbonate ($Na_2CO_3$). The solution was gently stirred for 20 minutes to develop color, ensuring minimal disturbance. The intensity of the color change was measured at 760 nm using a UV-Vis spectrophotometer

(Model: UV-1700 series). TPC was calculated based on the absorbance values. A final concentration of 0.1 mg/mL was used to evaluate both the extract and the standard samples [18]. Gallic acid was used as the reference standard to quantify TPC, expressed in milligrams of gallic acid equivalents (GAE) per gram of dry extract. The standard curve equation ($y = 0.0085$ $x + 0.1125$, $R^2 = 0.9985$) is shown in Fig 1.

## 2.9. Determination of total flavonoid content (TFC)

The aluminum chloride ($AlCl_3$) colorimetric method was utilized to ascertain the total flavonoid content of the crude methanolic extract and its partitionates, which include PSF, MSF, ESF, CTS, and CSF [19]. To summarize, 0.1 mL of 10% $AlCl_3$ and 1.5 mL of crude extract ($3.12 \times 10^{-2}$ mol/L methanol) were combined, and 0.1 mL of 1 mol/L Na-acetate was then added to the reaction mixture. The mixture was left to stand for half an hour. Next, 1 mL of a 1 mol/L NaOH solution was added, and the combination was finally brought to a final volume of 5 mL using double-distilled water. After the mixture was let to stand for 15 minutes, the absorbance at 415 nm was determined. A calibration curve was used to determine the total flavonoid content. Catechin was used as the standard to quantify TFCs as mg of CE/g of dry extract, the CE equivalent. The total flavonoid content was ascertained using the calibration curve outlined in Fig 1.

$Y = 0.01X + 0.0409$, $R^2 = 0.9921$, where X is the catechin equivalent and Y is the crude extract absorbance, represented.

## 2.10. Approval for ethics

This study was carried out in strict accordance with the recommendations in the Guide for the Care and Use International Centre for Diarrheal Disease Research, Bangladesh (ICDDRB) guidelines, which was followed for conducting animal experiments. The World University of Bangladesh Ethical Committee Ethics of Animal Experiments read out the study through and gave its approval (Approval no # WUB/2023/242L6). The Ethical Committee on Animal Experiments at the World University of Bangladesh, in collaboration with our research team, mandated that the experimental mice used in this study would not be reused for any subsequent experiments. Upon completion of the study, all were promptly and humanely euthanized to minimize distress, fully in accordance with ethical guidelines. We strictly adhered to humane euthanasia methods and environmentally responsible handling practices, ensuring that we met the highest standards of animal welfare and ethical responsibility.

## 2.11. Experimental animal care

For this experiment, 6–8-week-old Swiss albino mice, weighing 40–45 gram on average were employed. Mice were housed under controlled environmental conditions, with a temperature maintained between 20 °C and 24 °C, relative humidity at 30%, and a photoperiod of 12 hours of light followed by 12 hours of darkness. The animal rooms and cages were regularly cleaned, and suitable nesting material was provided within the cages to enable the mice to regulate their microclimate effectively, in addition allow them to perform their natural nesting behaviors. They were given Jahangir Nagor University, Bangladesh -formulated foodad-libitumand tap water. Mice kept in a typical setting for a week at the World University of Bangladesh research laboratory to help them acclimate.

## 2.12. Method of sacrifice

The method of cervical dislocation for euthanizing mice under 200 grams was performed without anesthesia, adhering to the 2020 AVMA and IACUC guidelines [20]. Trained

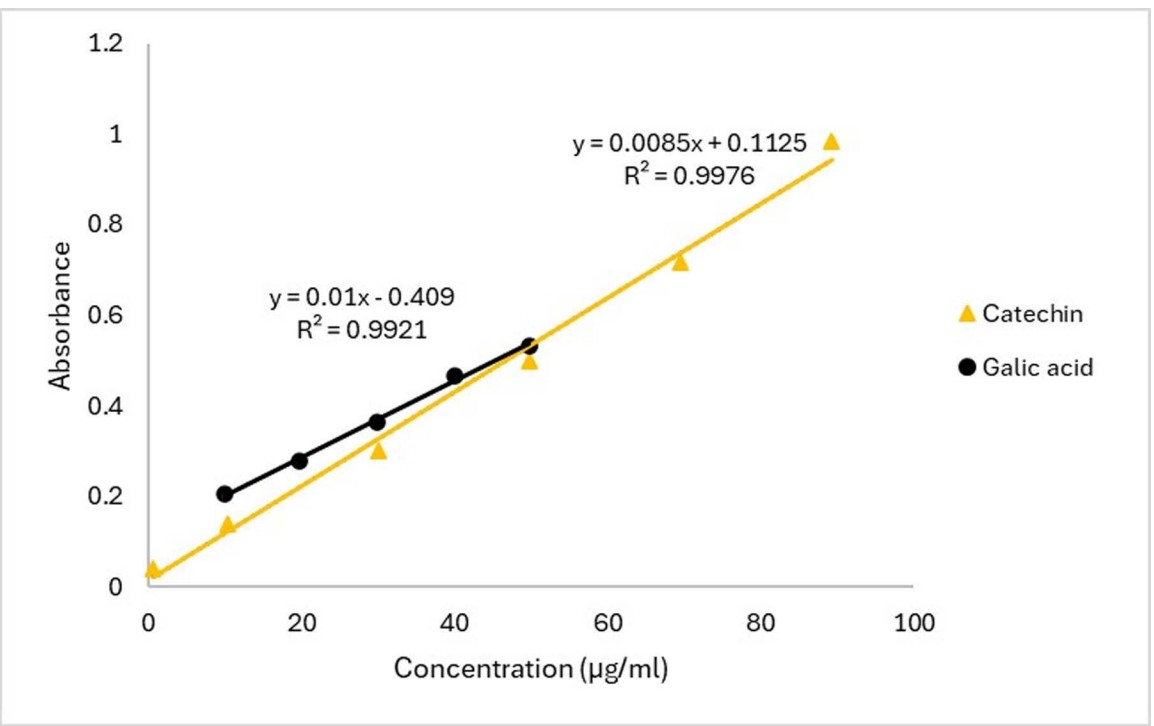

**Fig 1. Calibration curve of catechin and gallic acid.**

personnel carried out the procedure skillfully, using just the right amount of force to ensure the dislocation happened quickly and effectively, minimizing any pain or distress. During cervical dislocation, the thumb and index finger were positioned on either side of the animal's neck at the base of the skull while the animal lay on a table surface. On the other hand, the base of the tail is firmly and steadily pulled to cause separation of the cervical vertebrae and spinal cord from the skull. Following the procedure, we monitored the mice for over 60 seconds until the heartbeat ceased before disposal. Trained personnel monitored the mice for signs of pain or distress throughout the process. Critical assessment of the presence or absence of pain and/or distress between control and drug treated mice behavior has been taken carefully. To prevent any environmental contamination, the carcasses were wrapped in a plastic bag, placed in a box, and buried underground. When choosing a burial site, we made sure there were no nearby underground water sources or flood-prone areas to prevent water contamination. We also looked for soil that wasn't too sandy [21].

## 2.13. Acute toxicity test

The acute toxicity test was conducted according to the OECD (Organization for Economic Co-operation and Development) Test No. 423: Acute Oral Toxicity Acute Toxic Class Method (2002), with minor modifications, to evaluate the acute toxicity of *Litsea glutinosa* methanol leaves extracts [22]. The extract was administered carefully to mice using a feeding needle by gavage or via the intraperitoneal route at varying concentrations: 200, 400, 800, 1600, and 3200 mg/kg [23].

Before administering the extract, all mice were fasted for 16 hours. After treatment, the mice (six per group) were observed continuously for 1 hour, then intermittently over the next 4 hours, and finally monitored for a 24-hour period to detect any behavioral changes, signs

of toxicity, or symptoms of mortality. The animals were further observed for latency to death over a 14-day period.

Throughout the experiment, the mice had unrestricted access to food and water. Observations regarding acute toxicity signs, including changes in daily food intake, water consumption, body weight, and mortality, were recorded over a 48-hour period [24]. The $LD_{50}$ values were determined using the method described by Litchfield and Wilcoxon [25].

## 2.14. Analgesic activity test

In the current investigation two different methods were employed for testing the possible peripheral and central analgesic effects of *L. glutinosa* leaves; namely acetic acid induced writhing test and hot plate test in mice respectively.

### 2.14.1. Acetic acid induced writhing in mice.

The analgesic activity of the crude methanolic extract of *Litsea glutinosa* leaves was evaluated using the acetic acid-induced writhing model in mice, as described by Koster *et* al. [26]. The animals were divided into four groups: negative control (Group I), positive control (Group II), and two test groups (Groups III and IV). The test groups received LG extracts at doses of 250 mg/kg (Group III) and 500 mg/kg (Group IV) body weight, respectively. The positive control group was administered diclofenac (standard drug) at a dose of 25 mg/kg body weight, while the negative control group was treated with 1% Tween 80 in distilled water at a dose of 10 mL/kg body weight.

Test samples, standard drugs, and control vehicles were administered orally 30 minutes before the intraperitoneal injection of 0.7% acetic acid. Fifteen minutes after the acetic acid injection, the writhing responses (characterized by abdominal constrictions, trunk twisting, and hind leg extension) were observed in the mice for 5 minutes. Acetic acid activates pain receptors and induces writhing by releasing endogenous chemicals [27]. The number of writhes (squirms) exhibited by each mouse was meticulously counted for 15 minutes. The percentage inhibition of writhes was calculated using the following formula:

$$Percentage\ inhibition = \left\{ \left[ \begin{array}{l} Average\ number\ of\ writhes\ for\ control \\ -\ Average\ number\ of\ writhes\ for\ test \\ \div Average\ number\ of\ writhes\ for\ control \end{array} \right] \right\} \times 100\%$$

### 2.14.2. Hot plate test in mice.

The method described by Eddy and Leimbach was used to assess the centrally acting analgesic activity of the *Litsea glutinosa* leaves extract [28]. The hot plate temperature was maintained at 45 ± 0.5 °C to induce a nociceptive pain response, characterized by behaviors such as paw licking (fore or hind paws) or jumping. Only animals that responded to the hot plate within 30 seconds were selected for the experiment.

Thirty mice were randomly divided into five groups, each consisting of six mice. Group I (negative control) received distilled water (10 mL/kg), while Group II (positive control) was treated with morphine (5 mg/kg). Groups III and IV received the crude methanolic extract of LG at doses of 250 mg/kg and 500 mg/kg, respectively. To prevent paw damage, a cut-off time of 20 seconds was established for the test. One-hour post-treatment, the reaction time (the time taken by the mice to jump, lick, or flutter their paws) was recorded at 0, 30, 60, 90, and 120 minutes.

## 2.15. Antipyretic activity test

The antipyretic activity of the crude methanolic extract of LG leaves was evaluated using a Brewer's yeast-induced pyrexia model in experimental animals [29]. Hyperpyrexia was induced by the subcutaneous injection of 20% aqueous brewer's yeast suspension at a dose

of 10 mL/kg body weight. The animals were fasted overnight with free access to water before the experiment. The initial rectal temperature of each animal was measured using an Ellab thermometer and recorded as 33.19 ± 0.40 °C.

Eighteen hours after the yeast injection, animals that exhibited a rectal temperature increase of 0.3–0.5 °C were selected for the antipyretic activity study. The selected animals were orally administered the crude methanolic extract of LG at doses of 100, 200, and 300 mg/kg. Paracetamol (100 mg/kg) served as the reference drug, while the negative control group received only distilled water (10 mL/kg). Rectal temperatures were recorded at intervals for up to 3 hours post-treatment [30].

## 2.16. Test organisms for antimicrobial studies

Five Gram-positive and five Gram-negative bacteria were used in the antibacterial screening of five organic soluble fractions of *Litsea glutinosa* leaves. The bacterial cultures included *Actinomyces, Bacillus subtilis, Bacillus cereus, Staphylococcus aureus, Sarcina lutea, Salmonella typhi, Chlamydia trachomatis, Escherichia coli, Vibrio mimicus, and Bacillus parahemolyticus*. The cultures were maintained at 37 °C. Bacterial isolates were obtained from the Microbiology Department at Dhaka University.

**Preparation of inoculum.** To prepare the bacterial inoculum, 5 mL of nutrient broth was pipetted into each test tube, sealed, autoclaved, and allowed to cool. The test tubes were labeled and arranged on a rack. Two loopfuls of each bacterial test strain were added to the broth, mixed thoroughly, and incubated at 37°C for 2 hours. After incubation, visible growth of the organisms was observed.

**Antibacterial activity.** The antibacterial activity of *L. glutinosa* leaves extracts was evaluated using the disc diffusion method. The antimicrobial properties of the organic fractions of leaves extract included the carbon tetrachloride fraction (CTF), petroleum ether fraction (PSF), ethyl acetate fraction (ESF), methanol soluble fraction (MSF), and chloroform fraction (CSF). Their antimicrobial activity was compared to that of a standard antibiotic disc (Ciprofloxacin, 5 μg/disc) and methanol, which was used as a negative control. All tests were conducted in triplicate, and the results were reported as mean ± SD. Agar media were prepared for culturing both Gram-positive and Gram-negative bacteria, including *Actinomyces, Bacillus subtilis, Bacillus cereus, Staphylococcus aureus, Sarcina lutea, Salmonella typhi, Chlamydia trachomatis, Escherichia coli, Vibrio mimicus*, and *Bacillus parahemolyticus.*

Sterile filter paper discs (6 mm in diameter) were impregnated with 400 μg of the test extract. After evenly seeding the pathogenic microorganisms onto the agar plates, the discs were placed on the surface of the agar. Nutrient agar was used to cultivate bacterial strains. Blank discs impregnated with 40 μL of methanol served as negative controls, while Ciprofloxacin (5 μg/disc) acted as the positive control. The plates were allowed to diffuse at 4 °C in a refrigerator for 24 hours, then incubated at 37 °C for another 24 hours. The antibacterial activity was assessed by measuring the diameter of the inhibitory zones surrounding the discs, which indicated the extent of bacterial growth inhibition [31].

## 2.17. Statistical evaluation

The experimental data was analyzed using the Statistical Package for the Social Sciences (SPSS) version 22.0 (SPSS Inc., Chicago, IL, USA). The results were determined by taking the average of three evaluations and displaying them as mean ± standard deviation (SD). Significant deviations (p-values < 0.05) between the means were determined by using the using the one-way ANOVA Duncan's Multiple Range test (DMRT) as post-hoc.

## 3. Results

### 3.1. Qualitative phytochemical analysis

The leaves of LG have been shown to contain various phytochemicals, including steroids, tannins, saponins, phenols, alkaloids, and flavonoids, as outlined in Table 1. These phytochemicals are known to play significant roles in the plant's biological activity. For instance, flavonoids and alkaloids have been associated with analgesic properties, potentially modulating pain pathways through interactions with key neurotransmitters or receptors involved in pain perception. Phenolic compounds are recognized for their antipyretic effects, likely mediated by their influence on the hypothalamic regulation of body temperature. Additionally, tannins and saponins have exhibited antibacterial activity against various pathogens, highlighting their potential role in the plant's defense mechanisms.

In this study, in addition to standard phytochemical screening tests, high-performance liquid chromatography (HPLC) was employed to confirm the presence and composition of these bioactive compounds in LG leaves extracts. The chromatographic profiles, as depicted in Fig 2a–2d, provide detailed insights into the specific compounds present, reinforcing the findings from the preliminary qualitative analyses.

To verify the presence of biologically active phytochemicals, *Litsea glutinosa* extract and its various organic-soluble fractions were subjected to multiple analytical tests. The phytochemicals identified in this study, including alkaloids, flavonoids, phenols, and saponins, are well-documented in the scientific literature for their therapeutic potential. These compounds are known to possess a range of beneficial properties, such as anti-inflammatory, antimicrobial, and antioxidant effects, which are widely recognized in the field of medical sciences. The findings from this investigation may provide valuable insights for the development of novel therapeutic agents, enhancing our understanding of *Litsea glutinosa* as a source of bioactive compounds with potential pharmaceutical applications [32].

### 3.2. Chemical standardization of HPLC

HPLC proves to be a highly robust technique for conducting comprehensive analyses of qualitative aspects of phyto compounds within plant extracts. In this study, we used gallic acid as the standard for phenolic compound dentification, while catechin functioned as the standard for identifying flavonoid compounds. The retention time for the phenolic compound standard (gallic acid) was 2.066, and the methanolic extract of LG exhibited a corresponding peak at 2.77 retention time presented in Fig 2a,b. Both the standard and the extract were measured using UVs pectrum at 270 nm, revealing the presence of gallic acid in the methanolic extract,

Table 1. **Preliminary phytochemical screening of crude methanolic extract of *Litsea glutinosa leaves*.**

| Phytochemicals | Name of the test | Remarks |
|---|---|---|
| Carbohydrates | Molisch's Test | ++ |
| Reducing sugars | Fehling solution test | + |
| Tannins | Lead acetate test | ++ |
| Alkaloids | Mayer' s test | ++ |
| Flavonoids | Shinoda test | ++ |
| Saponins | Frothing test | ++ |
| Steroids | Libermnn-Burchards test | + |
| Phenol | Ferric chloride test | ++ |

Abbreviations: Amounts shown as + = Mildly present, ++ = Moderately present, depending on the depth of color.

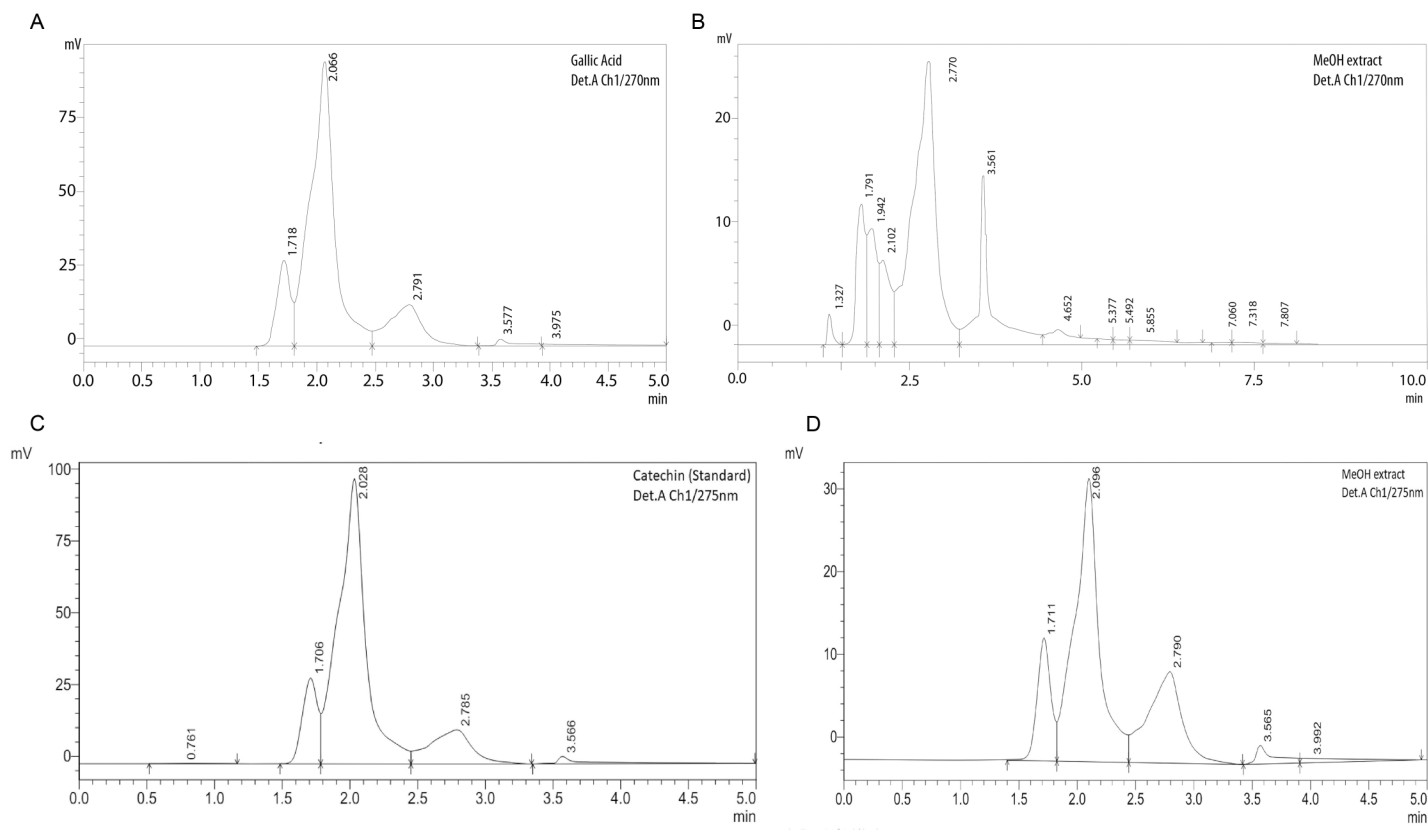

**Fig 2.** **(a) HPLC, Chromatogram for gallic acid standard. (b) Chromatogram of *Litsea glutinosa* leaves of methanolic extract. (c) HPLC, Chromatogram for cate-chin standard. (d) Chromatogram of *Litsea glutinosa* leaves of methanolic extract.**

indicating the presence of phenolic compounds. For the flavonoid compound standard (catechin), the retention time was 2.028, with the methanolic extract of *L. glutinosa* displaying a parallel peak at 2.09 retention time displayed in Fig 2c,d. UV spectrum measurements at 275 nm for both the standard and the extract indicated the presence of flavonoids in the methanolic extract. These findings line up with those of previous studies [33,34]. This outcome suggests that the methanolic extract of leaves contain both phenolic and flavonoid compounds, which are known for their analgesic, antipyretic and antibacterial activity, and other properties. Consequently, we assessed the total phenolic and flavonoid content in the plant extract and was considered a reasonable course of action.

### 3.3. Quantification of total flavonoid content

To quantify the catechin equivalent of total flavonoid content in the extract fractions, a complexometric method using aluminum chloride was applied. The flavonoid concentrations in the various organic extracts as MSF, PSF, ESF, CTF, and CSF were determined to be 234.30 ± 0.21, 131.01 ± 0.33, 233.20 ± 0.20, 239.31 ± 0.21, and 290.20 ± 0.12 mg of CE/g of dry extract, respectively, as shown in Table 2 and Fig 3a. Among these, CSF had the highest flavonoid content (290.20 ± 0.12 mg of CE/g of dry extract), while PSF had the lowest (131.01 ± 0.33 mg of CE/g of dry extract), followed by CTF, MSF, and ESF. The flavonoid concentration in CSF was significantly higher ($p < 0.05$). The substantial flavonoid levels observed in all extract fractions indicate that *Litsea glutinosa* leaves are a rich source of flavonoids, which may have

**Table 2. Total phenolic and total flavonoid contents of various extractives.**

| Sample | TPC (mg of GAE/g of extract) | TFC (mg of CE/g of extract) |
|---|---|---|
| MSF | 135.01 ± 0.57 | 234.30 ± 0.21 |
| PSF | 61.11 ± 0.55 | 131.01 ± 0.33 |
| ESF | 200.61 ± 0.47 | 233.20 ± 0.20 |
| CTF | 110.11 ± 0.37 | 239.31 ± 0.21 |
| CSF | 164.72 ± 0.15 | 290.20 ± 0.12 |

leave of *Litsea glutinosa*

Abbreviations: MSF, Crude methanol extract; PSF, Petroleum ether fraction; ESF, Ethyl acetate fraction; CTS, Carbon tetrachloride fraction; CSF, Chloroform fraction.

therapeutic potential for treating various illnesses. The results also demonstrate that the extraction efficiency for polyphenols and flavonoids is highly influenced by solvent polarity, as indicated by the variations in flavonoid content across the different solvents.

### 3.4. Quantification of total phenolic contents

Phenolic contents of organic soluble fractionates of *Litsea glutinosa* extractives were found to be 135.01 ± 0.57, 61.11 ± 0.55, 200.61 ± 0.47, 110.11 ± 0.37, and 164.72 ± 0.15 mg of GAE/g of MSF, PSF, ESF, CTF, CSF, in terms of GAE/g displayed in Table 2 and Fig 3b. The results indicated that the phenolic contents were highest in ESF (200.61 ± 0.47 mg of GAE/g of dry extract) and lowest in PSF (61.11 ± 0.55 mg of GAE/g of dry extract), with CSF and CTF coming in second and third. In comparison to MSF and PSF, the phenolic contents of ESF were meaningfully higher ($p < 0.05$). This finding implies that all the soluble fractions of *Litsea glutinosa* leaves contain phenolic chemicals to varied degrees. The study's findings indicate that leaves are a significant source of phenol with possible health advantages.

### 3.5. Acute toxicity

No signs of toxicity or deaths were observed after oral administration of crude methanol extract at concentration of 200, 400, 800, 1600, and 3200 mg/kg to evaluate the acute toxicity, within the first 48 hours, indicating that the oral $LD_{50}$ is more than above dose. Gross physical and behavioral examinations of the experimental mice revealed no noticeable acute poisoning symptoms, such as vomiting, diarrhea, or loss of appetite. ataxia, hypoactivity, piloerection, and syncope.

### 3.6. Analgesic activity(algia)

**Acetic acid-induced abdominal writhes in mice.** The analgesic activity of the methanolic extract of LG leaves was evaluated using the acetic acid-induced writhing test in mice. Acetic acid was used to induce pain by triggering the production of endogenous chemicals that activate pain-sensitive nerves, causing a writhing response in the mice.

In this experiment, the standard drug diclofenac and the methanolic extract of LG leaves were compared for their ability to reduce the writhing response. The plant extract at a dosage of 500 mg/kg body weight significantly reduced the number of writhing episodes in mice, demonstrating a 70% inhibition rate. In comparison, diclofenac at a dosage of 25 mg/kg body weight produced a 77.5% inhibition of writhing (as shown in Table 3).

Statistical analysis using a t-test (n = 5) revealed that both the diclofenac and LG leaves extract results were highly significant ($p < 0.001$). These findings suggest that the methanolic extract of LG leaves possesses a notable analgesic effect. However, further studies are

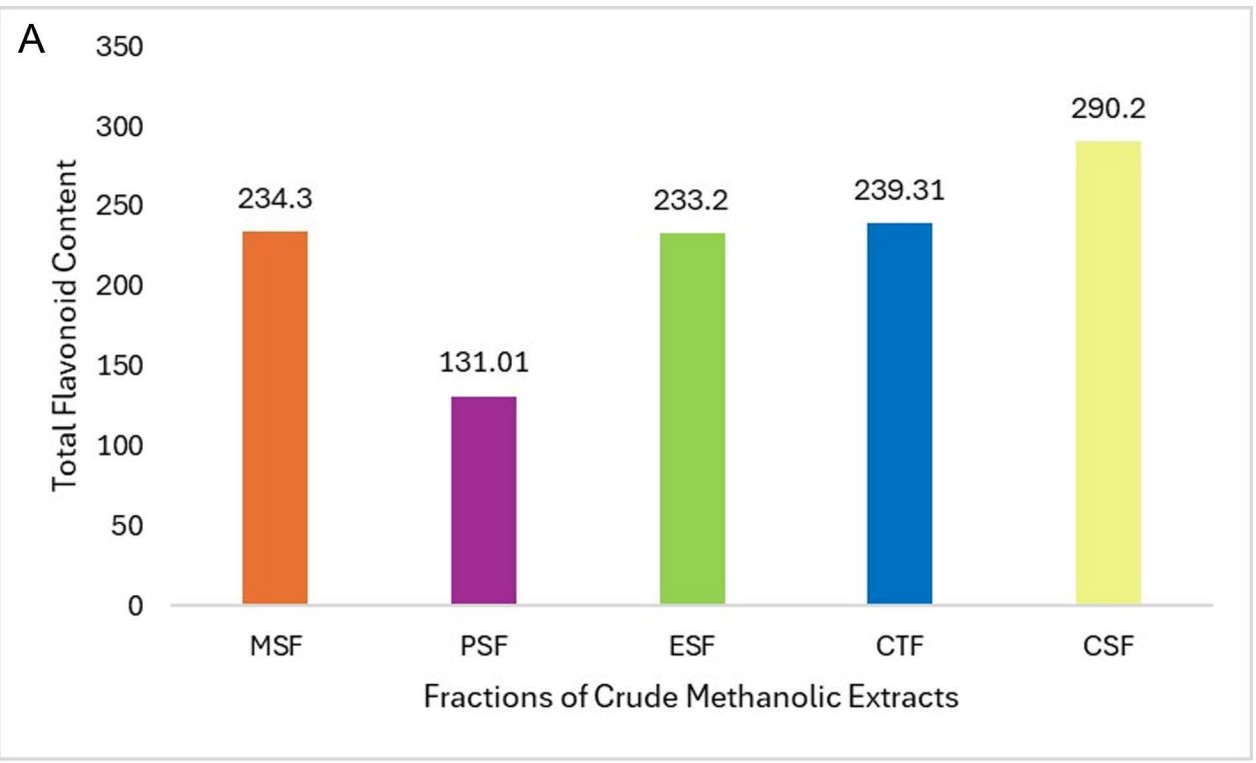

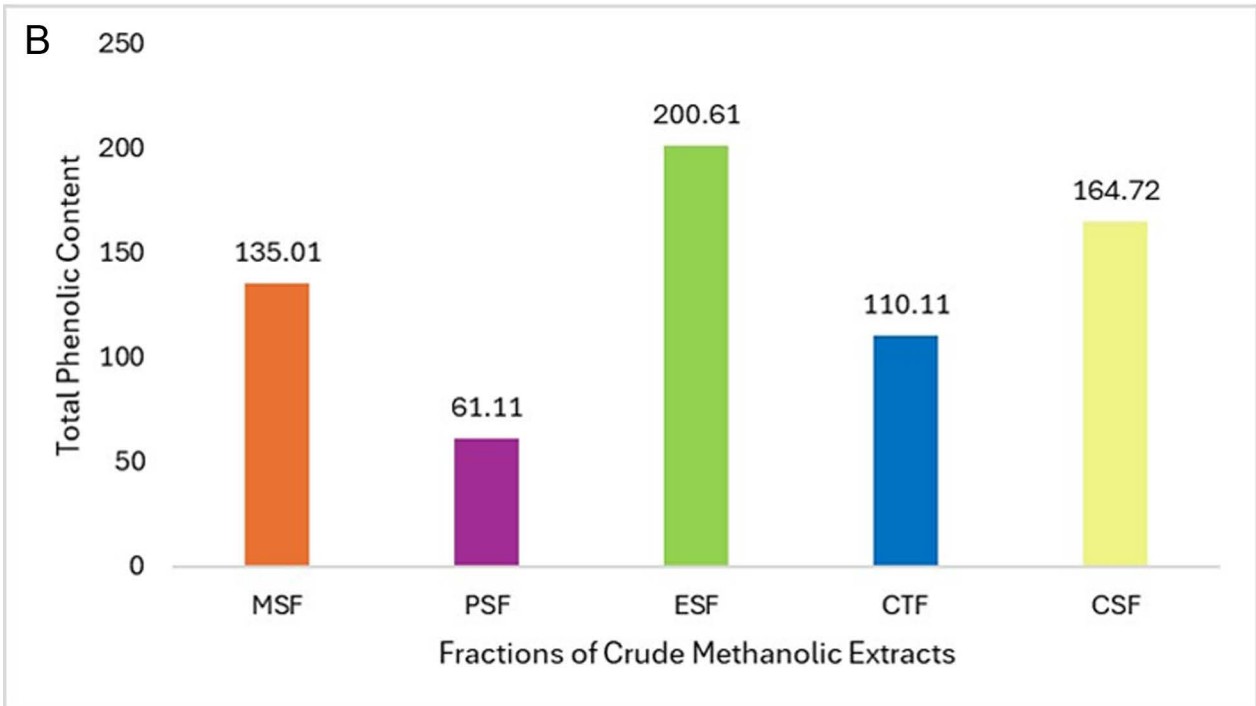

**Fig 3.** **(a) Total flavonoid content study** *in vitro* **evaluation of fractions of** *Litsea glutinosa* **leaves (b) Total phenolic content study of** *in vitro* **evaluation of fractions of** *Litsea glutinosa* **leaves.**

**Table 3. Statistical evaluation of *Litsea glutinosa* leaves extract on acetic acid induced writhing in mice.**

| Group | Number of writhing Mean ± SEM | % of Inhibition |
|---|---|---|
| Control (Tween 80) | 32 ± 1.22 | 0 |
| Diclofenac (25 mg/kg) | 7.2 ± 0.84 | 77.5 |
| Plant extract of LG (250 mg/kg) | 10.6 ± 1.00 | 68.1 |
| Plant extract of LG (500 mg/kg) | 9.6 ± 1.14 | 70 |

Abbreviations: Diclofenac as standard drug; LG is as *Litsea glutinosa* leaves and % inhibition = percent inhibition

**Table 4. Effect of methanolic leaves extract of *L. glutinosa* on hot plate test in mice.**

| Treatment (mg/kg) | Response time (sec) | | | | |
|---|---|---|---|---|---|
| | 0 min | 30 min | 60 min | 90 min | 120 min |
| DW 1 ml/kg; Group 1 | 1.63 ± 0.12 | 1.77 ± 0.13 | 2.17 ± 0.13 | 1.50 ± 0.10 | 1.47 ± 0.12 |
| Morphine (5 mg); Group II | 1.66 ± 0.22 | 5.17 ± 0.63 | 5.15 ± 0.27 | 6.45 ± 0.24 | 6.47 ± 0.23 |
| Extract LG (250 mg); Group III | 1.67 ± 0.21 | 2.17 ± 0.21 | 3.00 ± 0.17 | 3.00 ± 0.20 | 2.13 ± 0.20 |
| Extract LG (500 mg); Group IV | 1.51 ± 0.12 | 3.17 ± 0.17 | 3.90 ± 0.10 | 3.90 ± 0.37 | 3.37 ± 0.31 |

Abbreviations: DW = Distilled Water; LG = *L. glutinosa*

needed to isolate and identify the active compound(s) responsible for the analgesic properties observed in the methanolic extract of LG.

### 3.7. Hot plate test in mice

The methanolic extract of *L. glutinosa* leaves was tested for its analgesic activity using a thermal stimulus in Swiss albino mice. Reaction time to the thermal stimulus was measured at two dosages of the extract: 250 mg/kg and 500 mg/kg. The results, shown in Table 4, indicate that both doses of the extract significantly increased the reaction time to the thermal stimulus.

The highest pain inhibition was observed at the 500 mg/kg dosage of the crude extract, which exhibited a maximum reaction time of 3.37 ± 0.31 seconds. This result is comparable to the standard positive control drug, morphine, which showed a reaction time of 6.47 ± 0.23 seconds. The results were statistically significant ($P < 0.001$ and $P < 0.01$) when compared to both the positive control (morphine) and the negative control.

Values are presented as Mean ± SEM, and data were analyzed using repeated measures ANOVA followed by the Bonferroni post hoc test.

### 3.8. Antipyretic activity of the crude extract

The effects of different doses of *L. glutinosa* extracts on rectal temperature in yeast-induced febrile mice are summarized in Table 5. After 18 hours of yeast suspension injection, all mice developed fever, with rectal temperatures ranging from 38.03 ± 0.26 °C to 38.68 ± 0. 42 °C. Administration of the standard drug at a dose of 100 mg/kg body weight significantly lowered rectal temperature to 36.32 ± 0.67 °C after 3 hours. Similarly, the crude extract at 200 mg/kg resulted in a significant reduction in body temperature, reaching 36.17 ± 0.32 °C after 3 hours. This reduction in yeast-induced fever was statistically significant ($P < 0.01$) at 1, 2, and 3 hours when compared to the

**Table 5. Effect of the hydroalcoholic extracts of *L. glutinosa* on yeast-induced pyrexia in mice.**

| Treatment/Samples | Initial | 0.5 h | 1 h | 2 h | 3 h |
|---|---|---|---|---|---|
| Control (10 ml/kg) | 38.33 ± 0.44 | 38.70 ± 0.46 | 38.66 ± 0.44 | 38.50 ± 0.56 | 38.39 ± 0.66 |
| Paracetamol standard (100 mg/kg) | 38.28 ± 0.42 | 38.73 ± 0.55 | 38.80 ± 0.68 | 37.50 ± 0.63 | 36.32 ± 0.67 |
| LG Methanolic extract (100 mg/kg) | 38.80 ± 0.30 | 38.17 ± 0.40 | 38.90 ± 0.46 | 37.77 ± 0.25 | 36.60 ± 0.17 |
| LG Methanolic extract (200 mg/kg) | 38.69 ± 0.79 | 38.57 ± 0.25 | 38.27 ± 0.15 | 36.80 ± 0.36 | 36.87 ± 0.32 |
| LG Methanolic extract (300 mg/kg) | 38.43 ± 1.37 | 38.37 ± 0.60 | 38.45 ± 0.70 | 36.00 ± 0.21 | 36.17 ± 0.32 |

Control group received distilled water (10 ml/kg) only; LG as *L. glutinosa*

rectal temperature recorded at 0 hours. In the group treated with the 300 mg/kg dose, the extract also significantly reduced fever (P < 0.05) at 1, 2, and 3 hours compared to the 0-hour baseline. All tested doses of leave crude extract (100, 200, and 300 mg/kg) resulted in a substantial reduction in rectal temperature when compared to the control group, with statistical significance (P < 0.05 for all doses, and P < 0.01 for the 300 mg/kg dose).

### 3.9. Antibacterial activity

The antibacterial activity of five organic-soluble fractions from *L. glutinosa* leaves was evaluated against a panel of five Gram-positive and five Gram-negative bacterial strains. These strains included the Gram-positive *Actinomyces, Bacillus subtilis, Bacillus cereus, Staphylococcus aureus*, and *Sarcina lutea*, as well as the Gram-negative *Salmonella typhi, Escherichia coli, Shigella dysenteriae, Chlamydia trachomatis*, and *Vibrio mimicus*.

The tested fractions exhibited substantial antibacterial activity, with inhibition zones ranging from 11 mm to 22 mm, indicating effective bacterial growth suppression. These inhibition zones reflect the varying sensitivities of the bacterial cell walls to the different extractives.

The chloroform-soluble fraction (CTF), carbon tetrachloride-soluble fraction (CSF), and ethyl acetate-soluble fraction (ESF) demonstrated significantly greater antibacterial activity (p < 0.05) against both groups of bacteria. These fractions likely contain bioactive compounds such as flavonoids, phenolics, or alkaloids, which are known for their antimicrobial properties. Their ability to inhibit a wide range of bacteria, including both Gram-positive and Gram-negative strains, suggests that these extractives may interfere with essential bacterial processes such as protein synthesis, enzyme function, or nucleic acid metabolism. Table 6 presents the detailed results, showing the specific inhibition zones produced by each extractive. Notably, the extractives displayed more pronounced effects on *Bacillus cereus, Staphylococcus aureus*, and *Escherichia coli*, which are known to cause various infections in humans. This suggests that *L. glutinosa* could be a potential source of antibacterial agents for treating infections caused by these pathogens.

However, in this study, several extractives effectively inhibited both Gram-positive and Gram-negative bacteria, suggesting that the compounds in *L. glutinosa* possess mechanisms that can penetrate or disrupt this outer membrane, likely through disruption of lipid components or interference with cell wall biosynthesis.

## 4. Discussion

Polyphenols, which include flavonoids, phenolic acids, tannins, lignans, and coumarins, are naturally occurring secondary metabolites found in a wide range of plant tissues such as fruits, vegetables, cereals, roots, and leaves. Recent studies have underscored the significant biological functions of these compounds, particularly their role as antioxidants, where they mitigate damage caused by reactive oxygen species (ROS) and reduce the risk of

**Table 6. Antibacterial activity by different extractives of leaves of *L. glutinosa* through agar diffusion method.**

| Test bacteria | Inhibitory zone diameter (mm) | | | | | |
|---|---|---|---|---|---|---|
| | MSF | PSF | CTF | CSF | ESF | Che-5 |
| **Gram positive bacteria** | | | | | | |
| *Actinomyces* | 11.9 ± 0.21 | 12.5 ± 0.21 | 13.9 ± 0.21 | 14.5 ± 0.21 | 15.5 ± 0.21 | 35.0 ± 0.11 |
| *Bacillus subtilis* | – | 12.4 ± 0.45 | 13.5 ± 0.50 | 11.3 ± 0.15 | 11.9 ± 0.52 | 35.9 ± 0.96 |
| *Sarcinalutea* | 15.3 ± 0.42 | 17.2 ± 0.55 | 20.5 ± 0.57 | 22.2 ± 0.75 | 17.4 ± 0.45 | 33.4 ± 0.72 |
| *Staphylococcus aureus* | 13.5 ± 0.25 | 11.4 ± 0.31 | 19.5 ± 0.15 | 18.5 ± 0.45 | 19.4 ± 0.40 | 34.0 ± 0.15 |
| *Bacillus cereus* | 12.2 ± 0.25 | 11.5 ± 0.15 | 15.0 ± 0.32 | 19.4 ± 0.35 | 22.2 ± 0.81 | 33.0 ± 0.31 |
| **Gram negative bacteria** | | | | | | |
| *E. coli* | – | 13.5 ± 0.55 | 19.3 ± 0.52 | 19.2 ± 0.21 | 13.2 ± 0.52 | 32.0 ± 0.35 |
| *Shigella dysenteriae* | 11.5 ± 0.21 | 16.3 ± 0.15 | 17.5 ± 0.22 | 12.4 ± 0.87 | 17.4 ± 0.95 | 34.0 ± 0.15 |
| *Vibrio mimicus* | 11.5 ± 0.21 | 11.2 ± 0.25 | 11.3 ± 0.15 | 20.5 ± 0.91 | 16.0 ± 0.95 | 36.0 + 0.9 |
| *Chlamydia trachomatis* | 13.5 ± 0.32 | 11.9 ± 0.20 | 14.2 ± 0.35 | 16.2 ± 0.41 | 14.2 ± 0.86 | 33.0 + 0.91 |
| *Salmonella typhi* | 15.3 ± 0.17 | 12.9 ± 0.18 | 19.1 ± 0.85 | 12.2 ± 0.80 | 19.4 ± 0.64 | 30.0 + 0.05 |

Abbreviations: Che-5is Ciprofloxacin 5 µg/disc;(-) were not found.

oxidative stress-related pathologies. Phenolic compounds have drawn considerable scientific interest due to their diverse biological activities, including antioxidant, antimicrobial, anti-inflammatory, anticancer, and cardioprotective effects [35].

Flavonoids, a large and diverse subclass of polyphenolic compounds prevalent in many plant species, are recognized for their broad spectrum of biological actions. Among these, flavonoids exhibit antipyretic (fever-reducing) effects. Their antipyretic mechanisms are largely attributed to their anti-inflammatory properties, which involve the inhibition of pro-inflammatory mediators and modulation of the immune response [36]. Our study's quantitative analysis confirms that naturally occurring phytochemical compounds in *Litsea glutinosa* leaves contain satisfactory quantities of phenols and flavonoids, encouraging further evaluation of their pharmacological potential.

In the acetic acid-induced writhing test for peripheral analgesia in mice, pretreatment with 500 mg/kg b.w. of the methanolic extract significantly reduced the writhing frequency by 70% (p < 0.001), compared to 77.5% for the standard diclofenac 25 mg. Besides peripheral nociception, the LG extract also suppressed the central nervous pain response by significantly increasing the latencies and basal pain thresholds in the hot plate tests. The acetic acid-induced abdominal writhing test effectively screens compounds for peripherally and centrally acting analgesic activities [37]. Following intraperitoneal injection of acetic acid, endogenous chemicals such as PGE2, PGF2 α, PGI2, serotonin, histamine, lipoxygenase products, and peritoneal mast cells are released and accumulated, directly activating nociceptors and causing pain [38]. Mice react to this chemical stimulus by contracting their abdominal muscles, elongating their body parts, and extending their rear limbs. This response is believed to be mediated by local peritoneal receptors [39].

Aspirin and other NSAIDs reduce acetic acid-induced writhing by delaying the synthesis or release of these endogenous pain and inflammatory mediators [40]. The reduction in acetic acid-induced writhing by the leaves extract suggests peripherally mediated analgesic activity through blocking the synthesis or release of endogenous substances responsible for pain sensations. In the hot plate method, the extract significantly delayed the mice's reaction latency to thermally generated pain. However, morphine, the standard drug, more effectively inhibited the animals' thermal pain response than the leaves extract. Centrally acting analgesic compounds delay the response time to a heat stimulus [41]. Therefore, the leaves extract appears

to relieve pain by acting on the body's pain control systems, likely affecting both the brain and nervous system. It may activate opioid receptors in the brain and spinal cord, which are known to reduce pain. These receptors normally respond to natural pain-relieving chemicals in the body (like endorphins). By enhancing this system, the extract could prolong the time it takes for a person (or an animal) to feel pain, like how opioid painkillers work, but likely with fewer side effects. In addition to opioids, the extract might influence other chemicals in the brain that help calm down nerve activity, like GABA or somatostatin. These chemicals help reduce the transmission of pain signals, making the body less sensitive to pain. Many types of pain are linked to inflammation [42]. The leaf extract may reduce the release of inflammatory chemicals (like TNF-α and IL-6) that normally increase pain sensitivity. By lowering these inflammatory signals, the extract can help reduce both pain and swelling. The extract may also work by blocking COX enzymes, which are involved in producing prostaglandins that cause inflammation and pain. By stopping these enzymes, the extract can help reduce pain, much like how common painkillers (like ibuprofen) work.

The leaves extract seems to lower fever by interfering with the processes that trigger an increase in body temperature. When the body detects an infection or inflammation, it releases certain chemicals (like IL-1 and IL-6) that cause fever. The leaf extract might lower the levels of these chemicals, reducing the body's fever response.

Fever is mainly caused by PGE2, a substance that raises the body's temperature by acting on the brain's temperature control center (the hypothalamus). The extract may block the production of PGE2, helping to lower the fever by preventing this rise in temperature. The extract could also work by affecting the hypothalamus (the part of the brain that controls body temperature), helping the body cool down faster. It may reduce responses like shivering and blood vessel tightening (which increase body heat), while promoting sweating and blood vessel relaxation (which help cool the body [43].The bioactive compounds present in the extract, such as alkaloids, flavonoids, terpenoids, and polyphenols, are likely responsible for its pharmacological effects. These compounds are known to interact with key biological targets such as enzymes and receptors, potentially influencing various physiological processes. Based on our results, the extract exhibited analgesic activity by interacting with opioid receptors and modulating pain mediators like prostaglandins. Pharmacologically, this suggests inhibition of the cyclooxygenase (COX) pathway, specifically COX-2, which is responsible for the synthesis of prostaglandin E2 (PGE2), a key mediator of pain and inflammation. Additionally, the extract's ability to reduce fever may be linked to its suppression of pro-inflammatory cytokines, including interleukin-1 β (IL-1β), interleukin-6 (IL-6), and tumor necrosis factor-alpha (TNF-α). By inhibiting these cytokines and reducing PGE2 production, the extract effectively normalizes the hypothalamic set point, which contributes to its antipyretic (fever-reducing) effect [44].

The antibacterial studies demonstrated that a 400 μg dose of each extract effectively inhibited bacterial growth, producing inhibition zones ranging from 11 to 22 mm. Among the extractives, the chloroform-soluble fraction (CTF), carbon tetrachloride-soluble fraction (CSF), and ethyl acetate-soluble fraction (ESF) exhibited significantly larger inhibition zones (p < 0.05) against the tested bacterial strains compared to the standard antibiotic ciprofloxacin (5 μg/disc). Ciprofloxacin, a second-generation fluoroquinolone, is widely recognized for its broad-spectrum antibacterial activity against both Gram-positive and Gram-negative bacteria. It works by inhibiting bacterial DNA gyrase and topoisomerase IV, enzymes essential for DNA replication and transcription. Ciprofloxacin binds to bacterial DNA gyrase with 100 times higher affinity than to mammalian DNA gyrase, explaining its high efficacy against bacterial infections [45].

The results of the present study suggest that the extractives from *L. glutinosa*leaves, and their fractions possess notable antibacterial activity, potentially through mechanisms like those of ciprofloxacin, such as the inhibition of bacterial DNA gyrase and topoisomerase IV

[46]. These findings align with previous reports that demonstrate the effectiveness of organic solvent extracts of *L. glutinosa* against a wide variety of Gram-positive and Gram-negative bacterial strains.

Phenolic compounds, abundant in plant extracts, are considered key contributors to antibacterial activity. These compounds can disrupt bacterial cell membranes, interfere with critical metabolic processes, and ultimately induce cell lysis, leading to bacterial death. Moreover, certain phenolic compounds inhibit the formation of biofilms, complex structures that bacteria create to shield themselves from antibiotics and the host immune system. By preventing biofilm formation, phenolic compounds increase the efficacy of antibacterial treatments [47].

Furthermore, phenolic compounds can disrupt quorum sensing, a bacterial communication system that coordinates collective behaviors such as biofilm formation and virulence factor expression. By interfering with quorum sensing, these compounds reduce bacterial virulence and pathogenicity, making them less harmful to the host [32,48]. These multifaceted antibacterial mechanisms may explain the broad-spectrum activity observed in the *L. glutinosa* extracts.

In summary, the findings from this study support the hypothesis that *L. glutinosa* leaves extracts possess significant antibacterial properties, likely mediated by phenolic compounds and their ability to inhibit essential bacterial processes, such as DNA replication and biofilm formation. These extracts hold potential as a natural source of antibacterial agents, particularly in an era of increasing antibiotic resistance [49].

## 5. Conclusion

This study conducted phytochemical screening of methanolic crude extracts from *L. glutinosa* leaves and their soluble fractions, identifying compounds such as flavonoids, reducing sugars, tannins, gums, saponins, quinones, glycosides, steroids, and terpenoids. The extracts demonstrated significant analgesic, antipyretic, and antibacterial activities, supporting their traditional medicinal use. The research highlights the need for further investigation to isolate and identify the active components responsible for these effects. These findings suggest the potential of *L. glutinosa* as a source for developing new drugs and nutraceuticals, though additional studies, including clinical trials, are necessary to confirm these therapeutic applications. AcknowledgmentThe author gratefully acknowledges with sincere thanks, Department of Pharmacy of the World University of Bangladesh provided complete laboratory facilities for this research project. Each author has approved submission and taken full responsibility for the content of the work that has been submitted.

## Author contributions

**Conceptualization:** Zubair Khalid Labu, Samira Karim.

**Data curation:** Zubair Khalid Labu, Sarder Arifuzzaman.

**Formal analysis:** Samira Karim.

**Methodology:** Sarder Arifuzzaman.

**Software:** Md. Tarekur Rahman, Md. Imran Hossain, Md. Shakil.

**Supervision:** Md. Imran Hossain, Md. Shakil.

**Validation:** Md. Tarekur Rahman.

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
