## [Decision Letter · Decision Letter 0]

4 Oct 2024

PONE-D-24-35607

Assessment of Phytochemical Screening, Antibacterial, Analgesic, and Antipyretic Potentials of Litsea glutinosa (L.) Leaves Extracts in a Mice Model

PLOS ONE

Dear Dr. Labu,

Thank you for submitting your manuscript to PLOS ONE. After careful consideration, we feel that it has merit but does not fully meet PLOS ONE’s publication criteria as it currently stands. Therefore, we invite you to submit a revised version of the manuscript that addresses the points raised during the review process.

Please submit your revised manuscript before Nov 18 2024 11:59PM. If you will need more time than this to complete your revisions, please reply to this message or contact the journal office at plosone@plos.org . Please include the following items when submitting your revised manuscript:

We look forward to receiving your revised manuscript.

Kind regards,

Waqas Khan Kayani, PhD

Academic Editor

PLOS ONE

Journal Requirements:

3. We note that your Data Availability Statement is currently as follows: “All relevant data are within the manuscript and in Supporting Information files.”

Please confirm at this time whether or not your submission contains all raw data required to replicate the results of your study. Authors must share the “minimal data set” for their submission. PLOS defines the minimal data set to consist of the data required to replicate all study findings reported in the article, as well as related metadata and methods (https://journals.plos.org/plosone/s/data-availability#loc-minimal-data-set-definition). For example, authors should submit the following data: - The values behind the means, standard deviations and other measures reported; - The values used to build graphs; - The points extracted from images for analysis. Authors do not need to submit their entire data set if only a portion of the data was used in the reported study. If your submission does not contain these data, please either upload them as Supporting Information files or deposit them to a stable, public repository and provide us with the relevant URLs, DOIs, or accession numbers. For a list of recommended repositories, please see https://journals.plos.org/plosone/s/recommended-repositories. If there are ethical or legal restrictions on sharing a de-identified data set, please explain them in detail (e.g., data contain potentially sensitive information, data are owned by a third-party organization, etc.) and who has imposed them (e.g., an ethics committee). Please also provide contact information for a data access committee, ethics committee, or other institutional body to which data requests may be sent. If data are owned by a third party, please indicate how others may request data access.

Reviewers' comments:

Reviewer's Responses to Questions

**Comments to the Author**

1. Is the manuscript technically sound, and do the data support the conclusions?

Reviewer #1: Yes

Reviewer #2: Partly

2. Has the statistical analysis been performed appropriately and rigorously? 

Reviewer #1: Yes

Reviewer #2: No

3. Have the authors made all data underlying the findings in their manuscript fully available?

Reviewer #1: Yes

Reviewer #2: Yes

4. Is the manuscript presented in an intelligible fashion and written in standard English?

Reviewer #1: Yes

Reviewer #2: No

5. Review Comments to the Author

Reviewer #1: Manuscript is written scientifically well but need some minor revisions of grammatical mistakes as well formatting mistakes as some of them are mentioned below.

Unit for gram should be used consistently

Re-write line 187-188

Line-190; tape water should be replaced with tap water

In section 2.14 scientific names of bacteria should be italic

Figure number should be reviewed as fig 1a and 3A showed different format

In table 6, salmonella should be started with capital letter

The quality of all the pictures is not satisfactory, it should be improved

Reviewer #2: The article entitled, Assessment of Phytochemical Screening, Antibacterial, Analgesic, and Antipyretic Potentials of Litsea glutinosa (L.) Leaves Extracts in a Mice Model is interesting and different parameters are used to study, Antioxidant, Antibacterial, Analgesic, antipyretic phytochemical composition, including HPLC. Although the study is very unique and important as it describes the medicinal application of a plant extract against different bacterial strains and specially the use of Mice model. But few drawbacks are also there. Results are not presented with consistency. Methodology and Results section need to be improved because such a detailed study is not summarized properly.

Few points for to be improved are as:

1. Review of literature section is good however few grammatic and typo errors must be checked out. Recent articles should be included.

2. In Material and Methods section, the subheading Plant sample collection and accurate documentation should be “Sampling and proper documentation or Collection of Samples and Documentation.

3. It should be clear whether the plant material, only (Leaves) are collected for present study or Plants (Whole plant) was collected from field?

4. 3. Chemicals, reagents and instruments, the whole paragraph should be reviewed again and checked for few sentence structure errors. Similarly, Drying and grinding of plant materials and Extraction section should be checked and improved for final submission.

5. In results section lot of Typo errors are found and its difficult to mention all those errors one by one. Therefore, authors are suggested to review the article from a native English speaker again.

6. Discussion section should be improved after citing few recently published articles. Many articles about the plant used in the present study are available online.

Overall, the Results are interesting and methodology is appropriate, but incomplete and not written well. Figures including Chromatograms are of low quality. The authors are suggested to review the article in detail and encouraged to submit again after adding graphs and Figures with improved quality.

6. PLOS authors have the option to publish the peer review history of their article (what does this mean? ). If published, this will include your full peer review and any attached files.

**Do you want your identity to be public for this peer review?** For information about this choice, including consent withdrawal, please see our Privacy Policy .

Reviewer #1: No

Reviewer #2: No

---

## [Author Response · Author response to Decision Letter 0]

7 Nov 2024

Dear Dr. Kayani,

Thank you for the opportunity to revise my manuscript (ONE-D-24-35607) titled Assessment of Phytochemical Screening, Antibacterial, Analgesic, and Antipyretic Potentials of Litsea glutinosa (L.) Leaves Extracts in a Mice Model. I have carefully reviewed the feedback provided by the reviewers and made the necessary corrections to address all the points raised, including improvements in the manuscript's structure, language, and data presentation.

I have revised the manuscript accordingly and am now preparing the final documents for submission. I am optimistic that the revisions will meet the journal's standards, and I hope for a swift review and publication process.

Thank you once again for your guidance. I look forward to your further feedback.

Kind regards,

Zubair Khalid Labu

World University of Bangladesh

---

## [Decision Letter · Decision Letter 1]

16 Dec 2024

Assessment of Phytochemical Screening, Antibacterial, Analgesic, and Antipyretic Potentials of Litsea glutinosa (L.) Leaves Extracts in a Mice Model

PONE-D-24-35607R1

Dear Dr. Zubair Khalid Labu,

We’re pleased to inform you that your manuscript has been judged scientifically suitable for publication and will be formally accepted for publication once it meets all outstanding technical requirements.

Kind regards,

Waqas Khan Kayani, PhD

Academic Editor

PLOS ONE

Additional Editor Comments (optional):

Reviewers' comments:

Reviewer's Responses to Questions

**Comments to the Author**

1. If the authors have adequately addressed your comments raised in a previous round of review and you feel that this manuscript is now acceptable for publication, you may indicate that here to bypass the “Comments to the Author” section, enter your conflict of interest statement in the “Confidential to Editor” section, and submit your "Accept" recommendation.

Reviewer #1: All comments have been addressed

Reviewer #2: All comments have been addressed

2. Is the manuscript technically sound, and do the data support the conclusions?

Reviewer #1: Yes

Reviewer #2: Yes

3. Has the statistical analysis been performed appropriately and rigorously? 

Reviewer #1: Yes

Reviewer #2: Yes

4. Have the authors made all data underlying the findings in their manuscript fully available?

Reviewer #1: Yes

Reviewer #2: Yes

5. Is the manuscript presented in an intelligible fashion and written in standard English?

Reviewer #1: Yes

Reviewer #2: Yes

6. Review Comments to the Author

Reviewer #1: I am pleased to inform that I have reviewed the whole revised manuscript and found that you have made all suggested changes accordingly.

Reviewer #2: (No Response)

7. PLOS authors have the option to publish the peer review history of their article (what does this mean? ). If published, this will include your full peer review and any attached files.

**Do you want your identity to be public for this peer review?** For information about this choice, including consent withdrawal, please see our Privacy Policy .

Reviewer #1: No

Reviewer #2: No

---

## [Editor Report · Acceptance letter]

PONE-D-24-35607R1

PLOS ONE

Dear Dr. Labu,

I'm pleased to inform you that your manuscript has been deemed suitable for publication in PLOS ONE. Congratulations! Your manuscript is now being handed over to our production team.

Kind regards,

on behalf of

Dr. Waqas Khan Kayani

Academic Editor

PLOS ONE